# Limited gene flow and pronounced population genetic structure of Eastern Massasauga (*Sistrurus catenatus*) in a Midwestern prairie remnant

**Whitney J. B. Anthonysamy** [1] *****, **Michael J. Dreslik**[2], **Sarah J. Baker**[3], **Mark A. Davis** [2], **Marlis R. Douglas**[4], **Michael E. Douglas**[4], **Christopher A. Phillips**[2]

**1** Department of Basic Sciences, University of Health Sciences and Pharmacy in St. Louis, St. Louis, Missouri, United States of America, **2** Illinois Natural History Survey, Prairie Research Institute, University of Illinois Urbana-Champaign, Champaign, Illinois, United States of America, **3** Department of Biology, McNeese State University, Lake Charles, Louisiana, United States of America, **4** Department of Biological Sciences, University of Arkansas, Fayetteville, Arkansas, United States of America

* whitney.anthonysamy@uhsp.edu

**Data Availability Statement:** All relevant data are within the paper and its Supporting information files.

## Abstract

As anthropogenic changes continue to ecologically stress wildlife, obtaining measures of gene flow and genetic diversity are crucial for evaluating population trends and considering management and conservation strategies for small, imperiled populations. In our study, we conducted a molecular assessment to expand on previous work to elucidate patterns of diversity and connectivity in the remaining disjunct Eastern Massasauga Rattlesnake (*Sistrurus catenatus*) hibernacula in Illinois. We assayed genetic data for 327 samples collected during 1999–2015 from the Carlyle Lake study area across 21 microsatellite loci. We found hibernacula formed distinct genetic clusters corresponding to the three main study areas (Dam Recreation Areas, Eldon Hazlet State Park, and South Shore State Park). Genetic structuring and low estimates of dispersal indicated that connectivity among these study areas is limited and each is demographically independent. Hibernacula exhibited moderate levels of heterozygosity (0.60–0.73), but estimates of effective population size (5.2–41.0) were low and track census sizes generated via long-term mark-recapture data. Hibernacula at Carlyle Lake, which represent the only Eastern Massasauga remaining in Illinois, are vulnerable to future loss of genetic diversity through lack of gene flow as well as demographic and environmental stochastic processes. Our work highlights the need to include population-level genetic data in recovery planning and suggests that recovery efforts should focus on managing the three major study areas as separate conservation units in order to preserve and maintain long-term adaptive potential of these populations. Specific management goals should include improving connectivity among hibernacula, maintaining existing wet grassland habitat, and minimizing anthropogenic sources of mortality caused by habitat management (*e.g.*, mowing, prescribed fire) and recreational activities. Our molecular study provides additional details about demographic parameters and connectivity at Carlyle Lake that can be used to guide recovery of Eastern Massasauga in Illinois and throughout its range.

**Funding:** Funding for this project was provided through a grant between the University of Illinois Urbana-Champaign and the Illinois Department of Natural Resources (E-66-R-1; https://www2. illinois.gov/dnr/Pages/default.aspx) using United States Fish and Wildlife Services Section 6 funding that was awarded to WJBA, MJD, SJB, MAD and CAP. Funding was also provided, in terms of staff employment, through a contract between UIUC and the Illinois State Toll Highway Authority (RR-15-4228; https://www.illinoistollway.com/) that was awarded to MJD and CAP. In addition, this research was supported by two University of Arkansas Endowments: the Bruker Professorship in Life Sciences to MRD and the 21st Century Chair in Global Change Biology to MED (https://www.uark.edu/). The funders had no role in study design, data collection and analysis, decision to publish, or preparation of the manuscript.

**Competing interests:** The authors have declared that no competing interests exist.

## Introduction

In the Anthropocene, growing demands of human populations have altered natural habitats [1, 2] which exert ecological pressures on wildlife populations. Understanding how such pressures manifest in natural populations, particularly in declining or imperiled species, provides information essential for developing robust population models, adaptive management programs, and habitat conservation plans [3, 4]. A primary concern of declining populations is vulnerability to loss of genetic diversity through lack of gene flow, genetic drift and a subsequent reduction in effective population size and increased risk of inbreeding depression [5–7], which further accelerates populations into extinction vortices [8]. Here, molecular methods can complement traditional ecological approaches which evaluate population demographic and habitat use data to yield valuable insights into how anthropogenic pressures affect population viability and persistence of species in altered landscapes [9]. Molecular genetic data can elucidate population demographic parameters, population structure, landscape connectivity [10], and be used to assess loss of genetic diversity over time [11–13].

The Eastern Massasauga (*Sistrurus catenatus*) is a small cryptic rattlesnake formerly occupying a broad distributional area across a diversity of habitat types [14]. It ranges north-south from Ontario Canada to central Illinois and east-west from New York to the Mississippi River [15, 16]. Three geographic subunits have been identified; one for Iowa, Illinois, and Wisconsin, one for Indiana, Ohio, southern and central Michigan, and southwestern Ontario, and one for Pennsylvania, New York, northern Michigan, and other parts of Ontario, consistent with a northeastward post-Pleistocene range expansion from unglaciated into formerly glaciated regions ca. 10,000 years ago [17]. Contemporarily, the Eastern Massasauga exists in a fragmented landscape with most states having fewer than five extant populations, except for the Bruce Peninsula, Ontario, Canada, and Michigan which still maintain larger connected populations [18]. Primary threats include habitat fragmentation, loss, and modification, road mortality, hydrologic alteration, persecution, and habitat management practices such as mowing and prescribed fire [14]. Thus, isolation and barriers to movement among fragmented populations of Eastern Massasauga have potentially severed gene flow and increased genetic isolation, increasing the risk of genetic diversity loss. Consequently, the Eastern Massasauga is afforded some level of protection in every state or province where it occurs, and was formally listed as Threatened under the United States Endangered Species Act (ESA) in 2016 [19].

Range-wide molecular studies of Eastern Massasauga have revealed little phylogenetic variation [20], but notable population genetic structure and demographic independence over relatively small spatial scales, suggesting restricted contemporary gene flow among isolated populations within a fragmented landscape [21–24]. Further, Ochoa and Gibbs [25] found evidence for recent bottlenecks and increased inbreeding and Sovic *et al.* [11] predicted significant loss of genetic variation due to genetic drift in many populations over the next century. As anthropogenic threats continue to strengthen and increase dispersal barriers, understanding regional patterns of genetic structure, genetic diversity and connectivity of extant populations is necessary for evaluating long-term persistence, informing recovery plans, and identifying conservation units [26]. Given rapid population declines and ESA listing, studies examining the genetic composition of extant populations are essential to recovering the species.

In Illinois, Eastern Massasauga once extended throughout wet prairie habitats in the northern two-thirds of the state [27, 28]. However, populations declined with the conversion of prairie to agriculture; Illinois has lost 99.9% of prairie habitats since the industrial revolution [29]. At present, only one extant population remains, located at Carlyle Lake, Clinton County. During long-term studies of this last population, the top four mortality sources comprised

automobiles, predation, management-related (e.g. prescribed burns, vegetation control), and disease [30]. Within this region, most patches of habitat surveyed since 1999 contain fewer than 20 individuals [12, 31], yet anthropogenic activities continue to increase isolation and restrict gene flow [32, 33]. Previous molecular work based on microsatellite DNA analyses of Eastern Massasauga at Carlyle Lake revealed moderate to high levels of heterozygosity, but limited dispersal among sites [13, 21, 34]. These previous studies provide a foundation to investigate further population genetic structure at Carlyle Lake. However, samples used in these studies either did not represent the entire study area [13, 21], or used a limited number of loci [34], providing limited utility to recovery planning.

Our goal was to expand on previous genetic efforts by analyzing a broader temporal span, a spatial scale representative of the entire study area, and a greater number of microsatellite loci to provide a comprehensive molecular genetic assessment of Eastern Massasauga at Carlyle Lake, Illinois, that will be used to inform long-term conservation planning for the species. We sought to 1) investigate genetic diversity, structure and connectivity, and 2) determine spatial variation of allelic richness, effective population size, and level of inbreeding among primary study sites and remaining hibernacula at Carlyle Lake. Given the intense landscape conversion of the study area and historic and contemporary barriers to movement, we predicted to find distinct genetic clusters and limited gene flow among the hibernacula. Although previous studies showed moderate to high genetic diversity, we expected genetic indices to vary among hibernacula, with smaller and more isolated hibernacula exhibiting lower genetic diversity, lower effective population size, and higher levels of inbreeding. Our molecular study provides more in-depth information about population demographic parameters and connectivity at Carlyle Lake that can be combined with long-term ecological data [31, 35] to develop a comprehensive population model to guide recovery of Eastern Massasauga in Illinois. Further, as fragmentation and isolation are pronounced throughout the range of the Eastern Massasauga, these data can serve as a proxy for a range-wide perspective on the future of the species in the Anthropocene.

## Methods

### Study site

The U.S. Army Corps of Engineers constructed Carlyle Lake in 1961 in response to flooding issues in the middle Kaskaskia River Valley. The subsequent impoundment flooded much of the valley, completely submerging an expansive floodplain hardwood forest called Boulder Bottoms. The new impoundment effectively reduced or eliminated the east-west movements of many terrestrial organisms (Fig 1). A new channel was cut for the Kaskaskia River to accommodate the new spillway effectively separating the floodplain habitat west of the old channel (Dam East and West Recreation Areas; Fig 1). Much of the remaining natural habitat is concentrated along a thin band surrounding the lake in disjunct patches of public land. A map of the study area was generated from publicly accessible data downloaded from the Illinois Geospatial Data Clearinghouse [36] and from I-View [37] and then projected in ARCMAP 10.8.0 [38] (Fig 1). The public land is heavily used for outdoor recreational activities such as camping, fishing, hiking, and hunting. Beyond the band of more natural habitat lies a complex network of urbanization (roads and structures) and agriculture. Habitats range from upper terrace hardwood forests to ecotonal savannas to shrublands and grasslands. Managers use prescribed fire, mechanical and hand clearing, warm and cold season mowing, and herbicide applications to maintain habitats by controlling invasives and exotics. Eastern Massasauga occur within disjunct habitat patches isolated by roads, trails, and inhospitable habitats [32, 33].

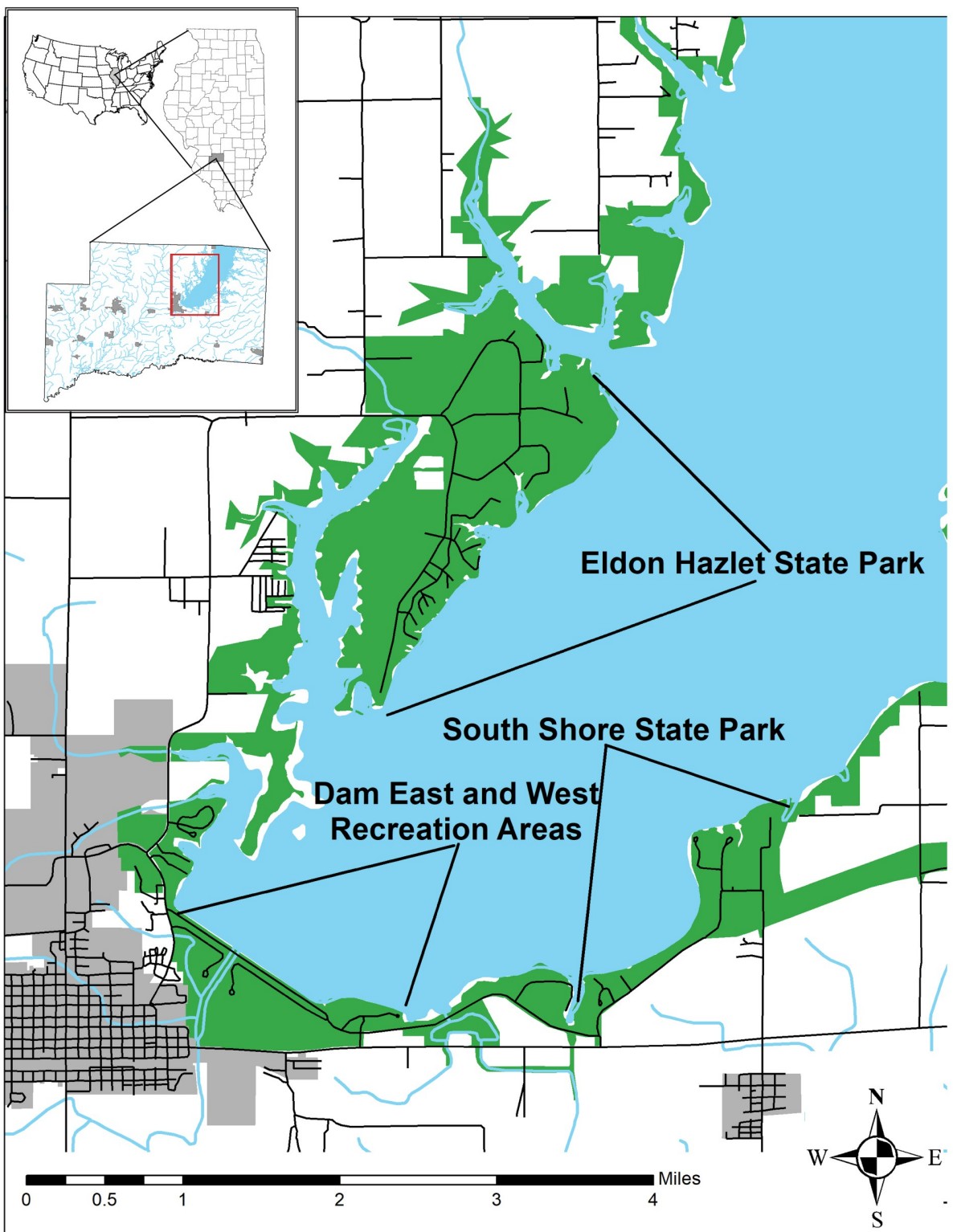

**Fig 1. Map of study site.** Sampling localities for 327 Eastern Massasauga collected from nine hibernacula across three study areas at Carlyle Lake, Illinois, USA. All map data (road layer, municipal boundaries, streams, lakes, and protected lands) were publicly accessible and downloaded from the Illinois Geospatial Data Clearinghouse [36] and from I-View [37] and then projected in ARCMAP 10.8.0 [38].

**Table 1. Standard genetic diversity indices for Eastern Massasauga Rattlesnake at Carlyle Lake.**

| Hibernacula | Code | N | #A | #PA | #RA | $A_R$ | $PA_R$ | $H_O$ | $F_{IS}$ |
|---|---|---|---|---|---|---|---|---|---|
| Dam East Recreation Area | DERA | 44 | 4.7 (0.5) | 6 | 33 | 3.51 (0.29) | 0.18 (0.08) | 0.63 (0.05) | -0.072 |
| Dam West Recreation Area | DWRA | 25 | 4.0 (0.3) | 1 | 21 | 3.43 (0.21) | 0.04 (0.02) | 0.60 (0.05) | -0.033 |
| Eldon Hazlet State Park—A | EHAR | 12 | 5.1 (0.5) | 0 | 42 | 4.53 (0.36) | 0.05 (0.05) | 0.73 (0.04) | -0.016 |
| Eldon Hazlet State Park—B | EHBR | 18 | 5.4 (0.6) | 1 | 43 | 4.42 (0.35) | 0.03 (0.02) | 0.70 (0.03) | 0.002 |
| Eldon Hazlet State Park—C | EHFT | 53 | 6.4 (0.6) | 3 | 61 | 4.41 (0.31) | 0.15 (0.06) | 0.72 (0.04) | -0.054 |
| Eldon Hazlet State Park—D | EHMD | 8 | 4.6 (0.4) | 0 | 32 | 4.44 (0.37) | 0.08 (0.05) | 0.66 (0.06) | -0.011 |
| Eldon Hazlet State Park—E | EHPR | 17 | 5.4 (0.5) | 2 | 43 | 4.44 (0.33) | 0.09 (0.06) | 0.66 (0.03) | 0.069 |
| Eldon Hazlet State Park—F | EHRR | 7 | 4.8 (0.5) | 0 | 32 | 4.76 (0.46) | 0.03 (0.02) | 0.72 (0.05) | -0.038 |
| South Shore State Park | SSSP | 143 | 6.7 (0.6) | 7 | 63 | 4.35 (0.32) | 0.23 (0.08) | 0.65 (0.04) | 0.030 |
| **Study Areas** | | | | | | | | | |
| Dam Recreational Area | DMRA | 69 | 5.6 (0.5) | 7 | 46 | 5.6 (0.5) | 0.3 (0.2) | 0.62 (0.05) | -0.028 |
| Eldon Hazlet State Park | EHSP | 115 | 7.1 (0.7) | 16 | 73 | 7.0 (0.6) | 0.8 (0.2) | 0.70 (0.03) | 0.013 |
| South Shore State Park | SSSP | 143 | 6.7 (0.6) | 7 | 63 | 6.3 (0.6) | 0.4 (0.1) | 0.65 (0.04) | 0.030 |

Provided are mean estimates for number of alleles ($^\#$A), number of private alleles (#PA), number of rare alleles (#RA), allelic richness ($A_R$), private allele richness ($PA_R$), observed heterozygosity ($H_O$), and inbreeding coefficients ($F_{IS}$) for 327 Eastern Massasauga sampled from nine hibernacula across three study areas as at Carlyle Lake, Illinois. Numbers in parentheses represent ±1 standard error. Data were derived from 21 microsatellite loci.

## Sampling

We included 392 samples collected from individuals captured during visual encounter surveys conducted during spring emergence from 1999–2015. Blood tissue was collected from the caudal vein and stored and -80 ˚C prior to DNA extraction. Samples represented nine hibernacula from the three major study areas (South Shore State Park, Eldon Hazlet State Park, and Dam Recreation Area) at Carlyle Lake (Fig 1; Table 1). The primary study areas are separated by distances of ~3–5 km and hibernacula (contiguous low-lying mesic grasslands with low canopy cover) within study areas are separated by unsuitable habitat and anthropogenic barriers that range from a few kilometers to a few hundred meters. Hibernacula locations are withheld and not provided on the study site map to protect the species from collection and persecution. All work on this project was conducted in accordance with Illinois Department of Natural Resources Endangered and Threatened Species Permit #05-11S and UIUC Institutional IACUC Protocol #14000.

## Laboratory protocols, genotype scoring, and data screening

Whole genomic DNA was extracted via Qiagen DNeasy Blood & Tissue Kit (QIAGEN Inc.) and assayed across 24 microsatellite loci [24, 39–43] using optimized PCR protocols [13]. Fragment analysis was carried out on an automated Applied Biosystems (ABI) GeneAnalyzer 3730xl at the W. M. Keck Center, University of Illinois, Champaign. An internal size standard (Liz 500) was included with each sample. Alleles were scored using GENEMAPPER 5.0 (ABI). Genotypes were compiled into a database and checked for possible null alleles and scoring errors using MICRO-CHECKER 2.2.3 [44]. Linkage disequilibrium was tested between all pairs of loci (Markov Chain parameters: 10000 dememorization steps, 500 batches, 5000 iterations) and departures from Hardy-Weinberg equilibrium (HWE) were evaluated for each locus using exact tests as implemented in GENEPOP 4.0 [45]. Levels of significance for multiple comparisons were adjusted using sequential Bonferroni correction [46]. Samples representing known offspring (N = 33) were not included in further analyses.

## Population structure and gene flow

Pairwise population structure was assessed using $F_{ST}$ analyses as well as the unbiased estimator $G''_{ST}$ [47] in GENALEX 6.5 [48, 49] to assess patterns of gene flow among hibernacula. Partitioning of genetic structure within and among hibernacula was evaluated using a hierarchical analysis of molecular variance (AMOVA) in ARLEQUIN 3.5 [50]. Numbers of distinct gene pools (K = 1–9) were assessed using a Bayesian clustering method implemented in program STRUCTURE 2.3.4 [51] using admixture ancestry and correlated allele frequency model parameter options for simulations (burn-in period = 500,000; iterations = 1,000,000; 10 repetitions). To determine the optimal number of clusters (*i.e.*, gene pools), we employed STRUCTURE SELECTOR [52] to calculate the *ad hoc* statistic ΔK [53] as well as the new estimators MedMeanK, Max-MeanK, MedMedK and MaxMedK which provide more reliable estimates of K when sampling is uneven [54]. Further, for the new estimators, we varied coefficient membership thresholds (0.5, 0.6, 0.7, 0.8) to explore how optimal K values changed across estimators [54]. We used CLUMPAK [55] to combine runs and visualize structure plots. Finally, rates of gene flow among gene pools over the last several generations were estimated using BAYESASS 3.0 [56].

## Genetic diversity

We estimated the standard diversity parameters of allele frequencies, rare and private alleles, and observed heterozygosity ($H_O$) using GENALEX 6.41 [48, 49] to characterize genetic diversity at each hibernaculum (and for each distinct gene pool). A rarefaction procedure, implemented in HP-RARE 1.0 [57], was used to correct for variable sample sizes in the calculation of allelic richness ($A_R$) and private allelic richness ($P_{AR}$). Inbreeding coefficients ($F_{IS}$) were calculated for each hibernaculum using FSTAT 2.9 [58]. Effective population size ($N_e$) and 95% CIs (jack-knife option) were estimated for each hibernaculum across three allele frequency exclusion values ($P_{crit}$ = 0.01, 0.02, 0.05) by using the linkage disequilibrium method implemented in LDNE 1.3 [59].

# Results

Of 359 samples, 32 were excluded because they either failed to amplify or could not be confidently genotyped (N = 17), represented duplicates (N = 9), or if mislabeling of field samples was suspected (N = 6). Of the 24 microsatellite loci, two were monomorphic (SCU-206, Scu-209), and a third (Scu-200) violated the assumptions of Hardy-Weinberg. In addition, three pairs of loci exhibited significant linkage disequilibrium for multiple hibernacula, but the exclusion of linked loci did not change results. Thus, the remaining 327 samples were evaluated across 21 microsatellite loci for the nine hibernacula and three major study areas (Table 1). Genotypes, locality abbreviation and year of collection are provided for each sample used in analyses (S1 File).

## Population structure and gene flow

Pairwise $F_{ST}$ and $G''_{ST}$ analyses were concordant and showed weak divergence among most of the hibernacula except the most proximate hibernacula within Eldon Hazlet State Park (Table 2). Patterns of divergence were stronger at a broader spatial evaluation among the three study areas (Table 3). When hibernacula were grouped by genetic cluster (EHSP, DMRA, SSSP), an Analysis of Molecular Variance (AMOVA) revealed that most genetic variation (87%) was partitioned within individuals, whereas only 8.7% and 4.2% were partitioned among genetic clusters and hibernacula, respectively (Table 4). STRUCTURE analyses indicated optimal K values ranging from 2–6 (Table 5, Fig 2). The ΔK estimator demonstrated the

**Table 2. Pairwise estimates of $F_{ST}$ (below diagonal) and $G''_{ST}$ (above diagonal) for 327 Eastern Massasauga sampled from nine hibernacula across three study areas as at Carlyle Lake, Illinois.**

| UNIT | DERA | DWRA | EHAR | EHBR | EHFT | EHMD | EHPR | EHRR | SSSP |
|---|---|---|---|---|---|---|---|---|---|
| DERA | - - - - | **0.141** | **0.365** | **0.384** | **0.449** | **0.441** | **0.371** | **0.414** | **0.366** |
| DWRA | **0.058** | - - - - | **0.309** | **0.352** | **0.407** | **0.346** | **0.354** | **0.337** | **0.386** |
| EHAR | **0.134** | **0.113** | - - - - | 0.076 | **0.129** | **0.092** | **0.132** | -0.029 | **0.312** |
| EHBR | **0.143** | **0.131** | 0.025 | - - - - | **0.106** | 0.126 | 0.052 | 0.104 | **0.322** |
| EHFT | **0.161** | **0.143** | 0.037 | 0.033 | - - - - | **0.231** | **0.181** | **0.151** | **0.422** |
| EHMD | **0.171** | **0.135** | 0.028 | 0.044 | **0.072** | - - - - | **0.174** | 0.076 | **0.337** |
| EHPR | **0.137** | **0.128** | **0.039** | 0.019 | **0.054** | **0.055** | - - - - | 0.138 | **0.335** |
| EHRR | **0.158** | **0.128** | -0.010 | 0.034 | **0.044** | 0.023 | 0.042 | - - - - | **0.290** |
| SSSP | **0.132** | **0.139** | **0.100** | **0.106** | **0.137** | **0.112** | **0.109** | **0.095** | - - - - |

Estimates were significant (in bold) at Bonferroni adjusted $P$-values alpha = 0.00138. Data were derived from 21 microsatellite loci, and full names of sampling locations are provided in Table 1.

**Table 3. Pairwise estimates of $F_{ST}$ (below diagonal) and $G''_{ST}$ (above diagonal) for Eastern Massasauga sampled from the three main study areas at Carlyle Lake, Illinois.**

| UNIT | DMRA | EHSP | SSSP |
|---|---|---|---|
| DMRA | - - - - | **0.345** | **0.351** |
| EHSP | **0.115** | - - - - | **0.336** |
| SSSP | **0.126** | **0.105** | - - - - |

Study areas corresponded to three gene pools as defined by Bayesian clustering analyses based on 327 Eastern Massasauga sampled from nine hibernacula. Estimates were significant (in bold) at Bonferroni adjusted $P$-values alpha = 0.01667. Data were derived from 21 microsatellite loci, and full names of sampling locations are provided in Table 1.

greatest rate of change at K = 2, clustering DMRA with EHSP. However, ΔK also indicated a high rate of change at K = 3, with the three genetic clusters corresponding to the three study areas (S1 Fig, S1 Table), though admixture of some individuals suggested dispersal among study areas. Given the tendency of the ΔK method to recover an inaccurate number of clusters and demonstrate a bias towards selecting K = 2 [60], particularly when sampling is uneven [54], we accepted K = 3 to be more biologically informative for our study and for conservation planning. For the new estimators (MedMeanK, MaxMeanK, MedMedK and MaxMedK) K values ranged from 3–6 though the number of optimal clusters tended to decrease as the

**Table 4. Analysis of Molecular Variance (AMOVA) for 327 Eastern Massasauga sampled from nine hibernacula across three study areas at Carlyle Lake, Illinois.**

| Source of Variation | d.f. | S.S. | Variance | % Variation | $P$-value |
|---|---|---|---|---|---|
| Among genetic clusters (study areas) | 2 | 389.6 | 0.69 | 8.73 | 0.004 |
| Among hibernacula within study areas | 6 | 118.1 | 0.33 | 4.22 | 0.000 |
| Among individuals within hibernacula | 318 | 2175.3 | 0 | 0 | 0.714 |
| Within individuals | 327 | 2263 | 6.92 | 87.6 | 0.000 |

Hibernacula were grouped by study area to test for genetic partitioning. Data were derived from 21 microsatellite loci.

**Table 5. Estimates of K values (genetic clusters) across all hibernacula.**

| Estimator | 0.5 | 0.6 | 0.7 | 0.8 |
|---|---|---|---|---|
| MedMedK | 5 | 5 | 5 | 4 |
| MedMeanK | 5 | 4 | 3 | 3 |
| MaxMedK | 6 | 6 | 6 | 6 |
| MaxMeanK | 6 | 6 | 5 | 3 |
| ΔK | 2 | | | |

Estimators MedMedK, MedMeanK, MaxMedK and MaxMeanK were calculated at four coefficient membership threshold values (0.5, 0.6, 0.7, 0.8).

threshold for coefficient membership increased (Table 5, S2 File). Visual inspection of the STRUCTURE plots revealed additional substructuring within the EHSP study area at K = 3 and K = 4, but structuring became less clear at K>4 (Fig 2). Further analysis of the EHSP study area supported additional substructure among hibernacula using the ΔK method (K = 2; Table 6, S2 Table) and the new estimators MedMeanK, MaxMeanK, MedMedK and Max-MedK (K = 1–3; S2 Fig, S3 File). At K = 2 and 3, the EHFT hibernaculum formed a distinct genetic cluster relative to the other hibernacula (Fig 3). A third gene pool emerged at K = 3 and consisted of 5–6 individuals from EHBR and EHRP. We note that observed structure is unlikely the result of temporal discordance among hibernacula or study areas because samples represented individuals collected throughout the duration of the study (1999–2015) and previous work found little change in genetic diversity over a 10-year period at SSSP [13].

When evaluating migration for the nine hibernacula with BAYESASS, MCMC for the simulations became trapped at the minimum bound (0.66) of the prior distribution. This is a common issue when population structure is weak [61]. Instead, as a more objective approach, proportions of migrants were estimated for the three primary genetic clusters (EHSP, DMRA, SSSP) delineated in the STRUCTURE analysis. Estimates of migration among clusters were low,

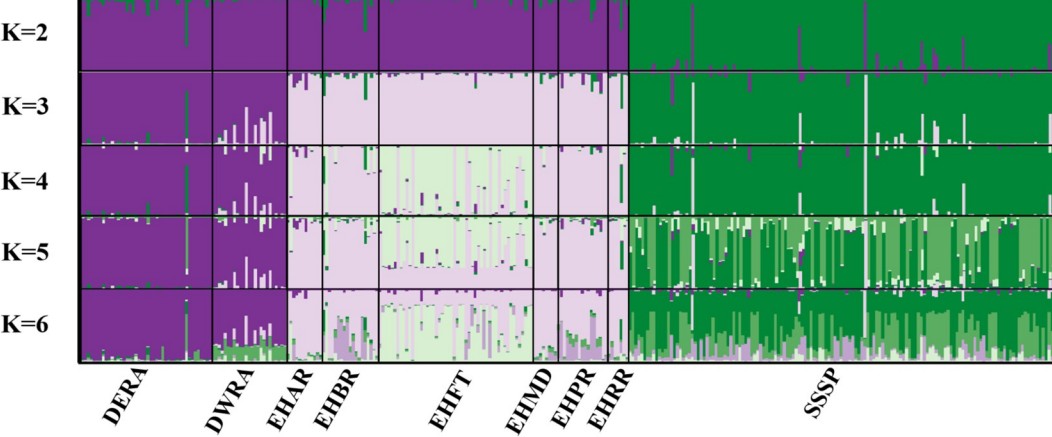

**Fig 2. Bayesian clustering results for 327 Eastern Massasauga sampled from nine hibernacula across three study areas at Carlyle Lake, Illinois.** STRUCTURE analyses revealed K = 3 distinct genetic clusters that corresponded to the three main study areas; Dam Recreation Area, Eldon Hazlet State Park, and South Shore State Park. Data were derived from 21 microsatellite loci, and full names of sampling locations are provided in Table 1.

**Table 6. Estimates of K values (genetic clusters) across Eldon Hazlet State Park (EHSP) hibernacula.**

| Estimator | 0.5 | 0.6 | 0.7 | 0.8 |
|---|---|---|---|---|
| MedMedK | 3 | 3 | 3 | 2 |
| MedMeanK | 3 | 2 | 2 | 1 |
| MaxMedK | 3 | 3 | 3 | 3 |
| MaxMeanK | 3 | 3 | 2 | 1 |
| ΔK | 2 | | | |

Estimators MedMedK, MedMeanK, MaxMedK and MaxMeanK were calculated at four coefficient membership threshold values (0.5, 0.6, 0.7, 0.8).

and the proportion of non-migrant individuals (>0.97) indicated demographic independence [62] for each study area (Fig 4).

## Genetic diversity

Hibernacula and study areas exhibited moderate levels of heterozygosity and low levels of inbreeding (Table 1). The Dam West Recreation Area (DWRA) demonstrated the lowest level of genetic diversity even though the hibernaculum was represented by the third largest sample size (N = 25). At a larger spatial scale, Eldon Hazlet State Park (EHSP) displayed slightly higher levels of genetic diversity than the Dam Recreation Area (DMRA) and South Shore State Park (SSSP). Across the nine hibernacula, 20 private alleles were detected, but this number increased to 30 when diversity was evaluated at the larger spatial scale (Table 1). Rare alleles were defined as those existing at ≤10% over the entire dataset and were detected for nearly all

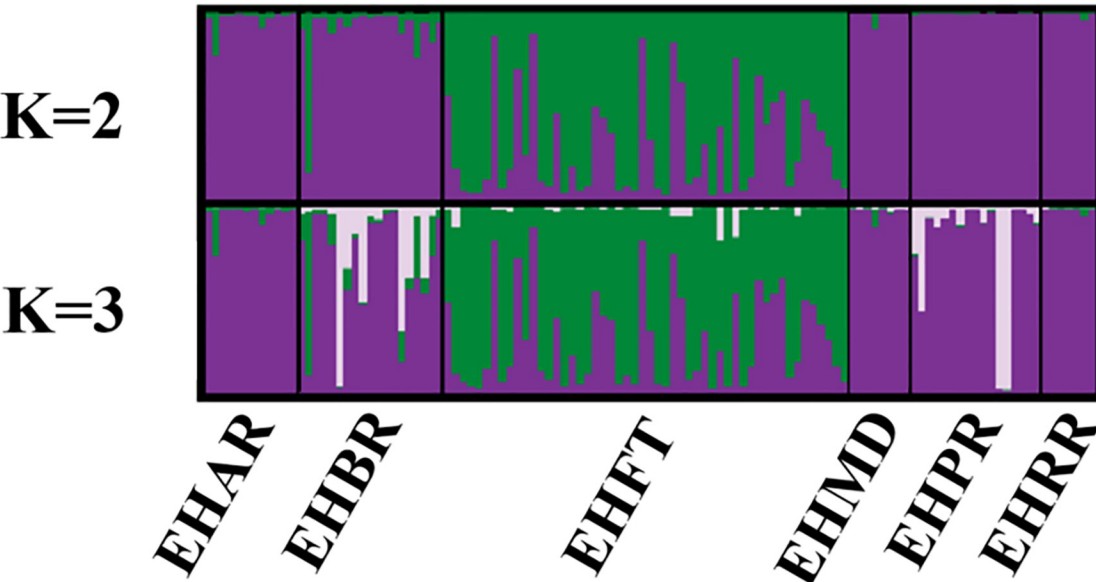

**Fig 3. Bayesian clustering results for 115 Eastern Massasauga sampled from six hibernacula within Eldon Hazlet State Park at Carlyle Lake, Illinois.** STRUCTURE analyses revealed K = 1–3 optimal genetic clusters. Data were derived from 21 microsatellite loci, and full names of sampling locations are provided in Table 1.

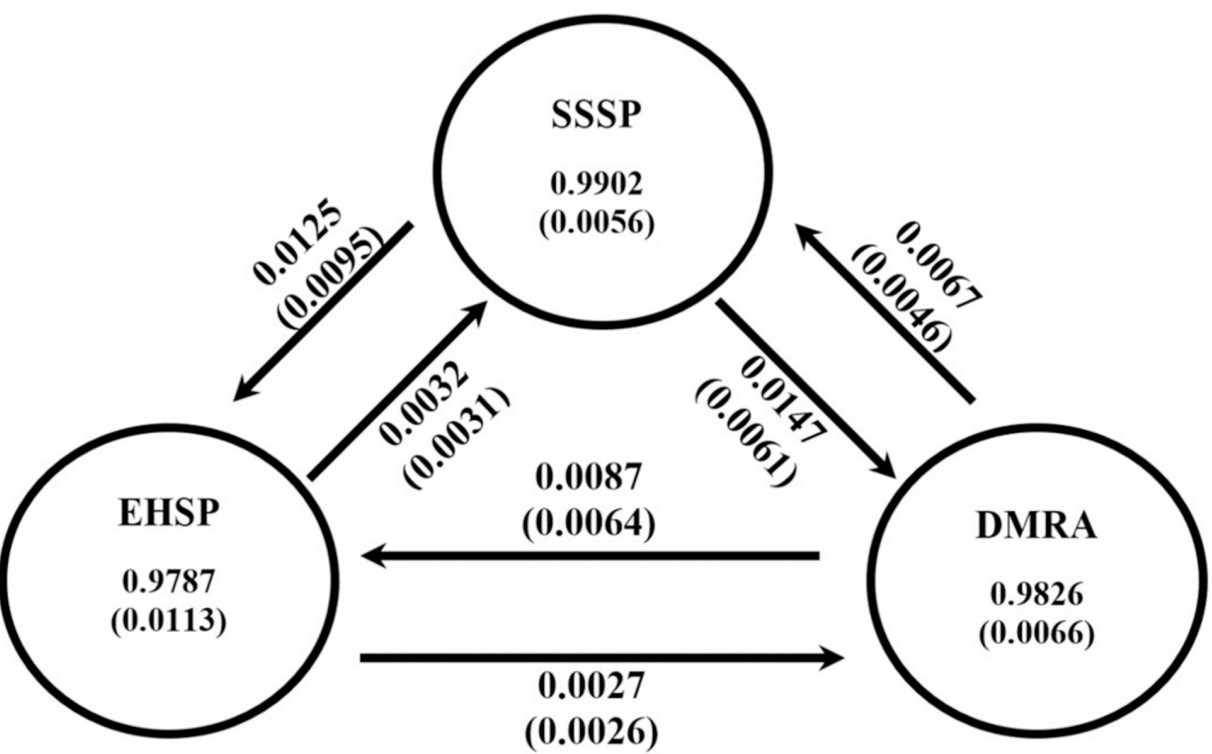

**Fig 4. Migration estimates for 327 Eastern Massasauga sampled from nine hibernacula across three study areas at Carlyle Lake, Illinois.** Structure analyses revealed K = 3 distinct genetic clusters that corresponded to the three main study areas; Dam Recreation Area (DMRA), Eldon Hazlet State Park (EHSP), and South Shore State Park (SSSP). Arrows represent estimates of asymmetric migration rates (±standard error) among the three genetic clusters, and circles represent estimates of the proportion of non-migrant individuals in each cluster. Data were derived from 21 microsatellite loci, and full names of sampling locations are provided in Table 1.

loci at every hibernaculum and study area, though numbers varied. Estimates of effective population size ($N_e$) were low and ranged from 5.2–41.0 among hibernacula and from 19.6–41.0 among the three genetic clusters (Table 7).

## Discussion

Our work expanded on previous genetic evaluations conducted across the range of the Eastern Massasauga, including Illinois [11, 21–24] and supported previous findings that habitat fragmentation and landscape barriers limit gene flow among remnant populations. However, our study at Carlyle Lake further demonstrated genetic structuring and lack of gene flow can occur not only among the primary study sites separated by ~3–5 km, but also among hibernacula separated only by a few hundred meters within the three primary study sites. We found support for at least three, but potentially four distinct gene pools at Carlyle Lake. South Shore State Park, Eldon Hazlet State Park, and the Dam Recreation Area all formed distinct genetic clusters, with further structuring within EHSP. Although we found moderate levels of genetic diversity across the hibernacula and little evidence for inbreeding, estimates of effective population size were low. These results are important for evaluating management of Eastern Massasauga at Carlyle Lake and recovery of the species in Illinois.

**Table 7. Estimates of effective population size ($N_e$) for Eastern Massasauga sampled from Carlyle Lake, Illinois.**

| | Critical Value | | |
|---|---|---|---|
| **Hibernaculum** | **0.05** | **0.02** | **0.01** |
| DERA | 21.1 | 30.6 | 25.2 |
| | (13.9–33.6) | (21.5–46.4) | (18.6–35.6) |
| DWRA | 7.4 | 7.8 | 7.8 |
| | (5.3–10.1) | (5.9–10.3) | (5.9–10.3) |
| EHAR | 17.3 | 32.9 | 32.9 |
| | (12.3–26.7) | (22.9–54.9) | (22.9–54.9) |
| EHBR | 6.8 | 11.3 | 11.3 |
| | (5.4–8.3) | (9.7–13.2) | (9.7–13.2) |
| EHFT | 18.4 | 21.8 | 23.8 |
| | (14.5–23.6) | (18.0–26.8) | (19.8–28.9) |
| EHMD | 8.6 | 8.6 | 8.6 |
| | (6.3–12.0) | (6.3–12.0) | (6.3–12.0) |
| EHPR | 5.2 | 7.9 | 7.9 |
| | (3.9–6.4) | (6.6–9.4) | (6.6–9.4) |
| EHRR | - - | - - | - - |
| SSSP | 32.1 | 38.4 | 41.0 |
| | (22.0–47.3) | (28.2–52.9) | (30.5–56.0) |
| **Study Area** | | | |
| DMRA | 19.6 | 22 | 23.6 |
| | (15.0–25.7) | (17.6–27.7) | (19.3–28.9) |
| EHSP | 28.4 | 30.9 | 36.5 |
| | (23.9–33.8) | (26.0–37.0) | (31.1–43.1) |
| SSSP | 32.1 | 38.4 | 41.0 |
| | (22.0–47.3) | (28.2–52.9) | (30.5–56.0) |

Estimates of $N_e$ and 95% CIs evaluated at three allele frequency exclusion values ($P_{crit}$ = 0.01, 0.02, 0.05) for 327 Eastern Massasauga sampled from nine hibernacula across three study areas at Carlyle Lake, Illinois. Data were derived from 21 microsatellite loci, and full names of sampling locations are provided in Table 1.

## Population structure and gene flow

Genetic structuring and limited connectivity among hibernacula at Carlyle Lake is likely attributed to both historical and anthropogenic disturbances [this study, 21, 25, 34] as well as high site fidelity and ecological requirements exhibited by the species [31]. The three main study areas (SSSP, EHSP, and DMRA) are all separated by natural and human-made landscape features including the Kaskaskia River, Carlyle Lake, paved roads, agriculture, and urbanization which impose barriers to dispersal [32, 33]. In our study, SSSP and EHSP, separated by 3.5 km of open water, showed strong genetic divergence, indicating populations on the east and west sides of the lake are genetically distinct [21]. Water bodies and roads had significant effects on gene flow in genetic simulations of regional Eastern Massasauga populations, but coexistence of natural and anthropogenic landscape features may confound inferences of contemporary effects on gene flow [63]. A number of effective migrants between SSSP and EHSP were low when evaluated within a historical and contemporary framework [21], suggesting population subdivision might predate construction of Carlyle Lake (*i.e.*, contemporary fragmentation), with the Kaskaskia River representing a historical barrier to dispersal and population

connectivity. Further, Ochoa and Gibbs [25] found that genomes of Eastern Massasauga (including SSSP) showed historically (~10,000 ybp) small $N_e$ estimates when compared to an outbred and non-threatened population of Western Massasauga (*S. tergeminus*), thus it could be argued that we would expect to find substantial genetic divergence among Eastern Massasauga populations that have been historically isolated. However, Ochoa and Gibbs [25] also found that Eastern Massasauga showed signals of recent (last 250 years) bottleneck events that correspond with anthropogenic fragmentation. Although we cannot directly measure historical levels of connectivity within the Carlyle Lake region or make reliable comparisons to undisturbed Eastern Massasauga populations using microsatellite DNA markers, recent genomic work highlights that anthropogenic landscape alteration and activities have significantly impacted population structure and gene flow of Eastern Massasauga throughout the species' range [25].

Genetic clustering and lack of connectivity are not unusual in studies of rattlesnakes [21, 64], and have shown genetic structuring among hibernacula in altered landscapes [65]. Rattlesnake populations occurring in anthropogenically disturbed habitats tend to display smaller home range sizes and higher site fidelity than those in undisturbed habitats which may be tied to availability of important habitats [66]. Ecological studies of Eastern Massasauga in the Midwest have shown similar patterns [67–69]. Eastern Massasauga exhibit high site fidelity and may be restricted by availability of hibernation sites [70]. Long-term telemetry and capture-mark-recapture studies at Carlyle Lake have shown little dispersal or movement among hibernacula [31]. Individuals monitored across several years maintained home ranges that averaged 4.3 ha (MCP) with maximum movements averaging ~160 m and the total length of movement paths averaging less than 2 km [35]. Additionally, most movements were localized with only males making larger forays during the breeding season and post-gravid females foraging for resources post-parturition [31, 71]. Overall, Eastern Massasauga at Carlyle slowly diffuse from and anchor movements around their overwintering sites [31]. Further, mark-recapture during visual encounter surveys showed little contemporary movement among hibernacula within sites and never between sites [31]. Like in other portions of their range, Eastern Massasauga at Carlyle Lake require crayfish burrows as overwintering refugia [31]. Thus, critical habitat needs may also explain high site fidelity at Carlyle Lake. These factors, in addition to landscape barriers, likely contribute to genetic structuring and lack of gene flow observed among the hibernacula at Carlyle Lake.

## Genetic diversity

Molecular approaches are important for understanding the impacts of anthropogenic disturbance on contemporary levels of genetic diversity, particularly in small, isolated populations that may be at risk of loss of genetic diversity, reduced effective population size and inbreeding effects [72]. Eastern Massasauga populations typically exhibit high to moderate levels of heterozygosity, even in small populations [73]. Previous microsatellite DNA studies at Carlyle Lake showed moderate to high levels of heterozygosity [21, 34] but no evidence of a long- or short-term genetic bottleneck at SSSP or EHSP [13, 21]. Further, in an evaluation of genetic diversity at the largest site, SSSP, observed heterozygosity ($H_o$) remained stable over the 10-year span, albeit a small, non-significant decrease in allelic diversity was noted [13]. Levels of inbreeding were non-significant, and there was no evidence of a recent population bottleneck. Genetic effective population size ($N_e$) ranged from 24–45 and was similar to the estimated census size ($N_c$) of 26–54 [13]. Range-wide analysis of genomic data also suggested low effective population sizes across the range of Eastern Massasauga for both, historical and contemporary time scales [25]. However, in contrast to studies using microsatellite DNA markers,

the recent genomic study of Ochoa and Gibbs [25] noted increased levels of inbreeding (FROH; fraction of genome covered by runs of homozygosity) in populations of Eastern Massasauga when compared to Western Massasauga. Historically low $N_e$ may have allowed Eastern Massasauga to purge highly deleterious mutations and exist in isolated populations while maintaining a moderate genetic load. Increased levels inbreeding in extant populations may reflect the relationship between the negative impacts of genetic drift in small populations and lack in selection against moderately deleterious mutations [25]. Moderate levels of inbreeding, even if the result of historical processes, are likely to threaten long-term persistence of Eastern Massasauga under the pressures of contemporary fragmentation. We also note that contemporary estimates of genetic diversity derived from microsatellite markers should be interpreted with caution, as it may take several generations for the loss of genetic diversity to be detected or manifest in small populations [11, 12]. Lack of inbreeding in our study suggests that microsatellite DNA markers offer limited utility in detection of inbreeding when $N_e$ is historically low and anthropogenic impacts are relatively recent. Nevertheless, small, disjunct populations often retain high levels of heterozygosity amid anthropogenic disturbances but may be at risk of loss of rare alleles, inbreeding and a decrease in effective population size in the future [74].

## Implications for recovery of Eastern Massasauga

Anthropogenic fragmentation can erode genetic diversity in vulnerable Eastern Massasauga populations as contemporary barriers to dispersal prevent gene flow, increase inbreeding depression, reduce adaptive variation, thereby accelerating populations into an extinction vortex [5, 7, 8]. Molecular genetic approaches provide insights into population-level processes in contemporary landscapes and thus promote recovery planning [75, 76]. Genetic data are essential for delimiting conservation units and understanding how anthropogenic factors, such as landscape changes or habitat management practices impede or facilitate gene flow [10, 77, 78]. The draft recovery plan [26] for Eastern Massasauga in the U.S. currently designates three conservation units (eastern, central, and western) based on haplotypes of a single mitochondrial DNA gene (ND2; [17]). While Ray *et al.* [17] considered samples across the range of Eastern Massasauga, providing valuable insight to the evolutionary history of the species, we argue that these conservation units are geographically broad, and do not address contemporary population-level processes or adaptive potential.

In addition to taxonomic distinctiveness, prioritization of recovery actions may need to vary across the species' geographic distribution depending on population-level genetic diversity and site-specific risks. There may be variation in stressors across the range, and within the three mtDNA-derived conservation units, that differentially affect survivorship and gene flow. For example, adult survivorship increases latitudinally in Eastern Massasauga [79]. Higher survivorship has been attributed to habitat management practices where anthropogenic influences are minimal [79, 80], though human-caused mortality including those caused by management practices often drive variation in survivorship patterns [79]. Eastern Massasauga at Carlyle Lake had the lowest annual adult survival (0.35) when compared to other populations across the latitudinal gradient, including those within the same conservation subunit [79], which could increase the risk of loss of future genetic diversity and rates of genetic drift compared to other populations. Further, recent work has shown evidence for differences in drift effects among populations for functional genetic variation that could impact adaptive variation in future generations [81, 82].

Eastern Massasauga at Carlyle Lake represent the remaining populations in Illinois and are located >500 km from the nearest extant populations (IA and WI) within the western conservation subunit [17]. In our study, we identified at least three distinct demographic units at

Carlyle Lake, with gene flow impeded among the larger study areas by major landscape barriers. Although contemporary genetic diversity remains high, these small populations are vulnerable to stochastic demographic and environmental processes that could decrease diversity, effective population size and adaptive potential in future generations [11]. Our work to evaluate genetic diversity and connectivity among hibernacula at Carlyle Lake highlights the need for recovery efforts to be tailored to a more regional, or local approach, when possible. Several conservation recommendations have been proposed to reduce ecological threats including seasonal closure of roads, mesopredator removal, and modifications to prescribed burn and mowing schedules to reduce incidental mortality [30, 32]. Steps to reduce mortality along with efforts to protect and restore habitat will be necessary to mitigate future population declines and preserve genetic diversity. Though evidence suggests that Eastern Massasauga has persisted historically at low effective population sizes and moderate genetic load [25], impacts of contemporary fragmentation and habitat loss warrant consideration of genetic rescue [83]. As inbreeding and genetic drift have been detected and predicted in current and future Eastern Massasauga populations, respectively [11, 25, 82], we also recommend, as part of a comprehensive recovery plan, consideration of genetic rescue efforts such as captive rearing and translocation to offset the impacts of contemporary fragmentation such as genetic drift, inbreeding and loss of adaptive potential [84]. Preliminary work on translocations of Eastern Massasauga have shown that short-distance (200 m) translocations can be successful, but that low overwinter survival is a challenge for long-distance translocation or augmentation from captive rearing [70, 85]. However, enrichment during captive rearing has been shown to improve reintroduction success in other snake species [86, 87]. In Illinois, individuals could potentially be moved among the primary study areas at Carlyle Lake to restore regional gene flow disrupted by contemporary fragmentation, though monitoring via radio-telemetry would be required to evaluate movement, body condition and survival of translocated individuals. Genetic rescue efforts should also consider donor individuals from populations with large $N_e$ that have minimal risk of introducing deleterious alleles while also maximizing adaptive potential [25, 83]. Though success of long-distance translocations are more challenging, for Eastern Massasauga at Carlyle Lake, genetic rescue may necessitate the admixture of individuals from a different conservation unit, such as those from larger, more intact populations found in Michigan and Canada.

Finally, genetic data is useful for incorporation into ecological data sets to aid in understanding the relationship between population trends, movement, habitat use and genetic drift and developing comprehensive population models to inform management decisions and recovery planning [88, 89]. Eastern Massasauga at Carlyle Lake have been consistently monitored since 1999 using visual encounter surveys, mark-recapture and radio-telemetry to estimate demographic parameters and characterize habitat use [31, 90]. The genetic data from this study can be combined with existing demographic estimates from long-term monitoring to develop quantitative population models that can help inform management of Eastern Massasauga at Carlyle Lake and guide recovery actions and criteria of the species across its range [91].

## Supporting information

**S1 Table. The rate in change (ΔK) between successive K values.** Estimates were derived using STRUCTURE SELECTOR [52] for 21 microsatellite markers across 327 Eastern Massasauga Rattlesnake individuals from Carlyle Lake, Illinois, USA.
(DOCX)

**S2 Table. The rate in change (ΔK) between successive K values for Eldon Hazlet State Park (EHSP).** Estimates were derived using Structure Selector [52] for 21 microsatellite markers across 115 Eastern Massasauga Rattlesnake individuals from Carlyle Lake, Illinois, USA. (DOCX)

**S1 Fig. Plot of ΔK values (K values 1–10) for Eastern Massasauga Rattlesnake sampled from Carlyle Lake, Illinois, USA.** (PDF)

**S2 Fig. Plot of ΔK values (K values 1–6) for Eastern Massasauga Rattlesnake sampled from Eldon Hazlet State Park (EHSP).** (PDF)

**S1 File. Sample information and microsatellite genotypes.** Locality abbreviation, sampling year and genotypes for 24 microsatellite markers across 327 Eastern Massasauga Rattlesnake individuals from Carlyle Lake, Illinois, USA. (XLSX)

**S2 File. Plots of K value estimates for MedMeanK, MaxMeanK, MedMedK and MaxMedK at the 0.5, 0.6, 0.7 and 0.8 thresholds for Eastern Massasauga Rattlesnake sampled from Carlyle Lake, Illinois, USA.** (PDF)

**S3 File. Plots of K value estimates for MedMeanK, MaxMeanK, MedMedK and MaxMedK at the 0.5, 0.6, 0.7 and 0.8 thresholds for Eastern Massasauga Rattlesnake sampled from Eldon Hazlet State Park (EHSP).** (PDF)

## Acknowledgments

Special thanks go to D. Shepard, B. Jellen, A. Stites, S. LaGrange, and D. Wylie for all their concerted efforts in the field. Thanks also go to USCOE Staff J. Smothers and D. Baum and IDNR staff J. Bunnell, K. Boyles, G. Tatham, J. Birdsell, S. Ballard, and M. Kemper for all their assistance with fieldwork and habitat management over the duration of the project. We thank J. M. Mui for creating the map figure.

## Author Contributions

**Conceptualization:** Whitney J. B. Anthonysamy, Michael J. Dreslik, Sarah J. Baker, Mark A. Davis, Marlis R. Douglas, Michael E. Douglas, Christopher A. Phillips.

**Data curation:** Whitney J. B. Anthonysamy, Michael J. Dreslik, Sarah J. Baker, Mark A. Davis, Christopher A. Phillips.

**Formal analysis:** Whitney J. B. Anthonysamy.

**Funding acquisition:** Whitney J. B. Anthonysamy, Michael J. Dreslik, Sarah J. Baker, Mark A. Davis, Marlis R. Douglas, Michael E. Douglas, Christopher A. Phillips.

**Investigation:** Whitney J. B. Anthonysamy, Michael J. Dreslik, Sarah J. Baker, Mark A. Davis, Marlis R. Douglas, Michael E. Douglas, Christopher A. Phillips.

**Methodology:** Whitney J. B. Anthonysamy, Michael J. Dreslik, Sarah J. Baker, Mark A. Davis, Marlis R. Douglas, Michael E. Douglas, Christopher A. Phillips.

**Project administration:** Whitney J. B. Anthonysamy, Michael J. Dreslik.

**Resources:** Marlis R. Douglas, Michael E. Douglas.

**Validation:** Whitney J. B. Anthonysamy.

**Visualization:** Whitney J. B. Anthonysamy, Michael J. Dreslik.

**Writing – original draft:** Whitney J. B. Anthonysamy, Michael J. Dreslik.

**Writing – review & editing:** Sarah J. Baker, Mark A. Davis, Marlis R. Douglas, Michael E. Douglas, Christopher A. Phillips.

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
