## [Decision Letter · Decision Letter 0]

19 Dec 2021

PONE-D-21-33512Limited gene flow and pronounced population genetic structure of Eastern Massasauga (Sistrurus catenatus) in a Midwestern prairie remnantPLOS ONE

Dear Dr. Anthonysamy,

Thank you for submitting your manuscript to PLOS ONE. Your work was assessed by 2 subject experts and myself. All 3 of us agree that the work represents an impressive major effort, and has the potential to be a very nice contribution to the conservation genetics literature. However, after careful consideration, we feel that it does not fully meet PLOS ONE’s publication criteria as it currently stands. Therefore, we invite you to submit a revised version of the manuscript that addresses the points raised during the review process.

REQUIRED CHANGES 1. Figure 1 needs to be updated to include a larger geographic context for the study site to reflect the broad international readership of PlosOne. 2. The authors need to address the concern raised during the review about recognizing contemporary vs. historical genetic differentiation, and provide some more specific guidance to readers on the importance of anthropogenic habitat fragmentation. It is not necessarily required to do any additional analyses (the reviewer suggests a simulation model for ideal circumstances as a comparison), the authors may be able to handle this with some strategic revisions to the text. 3. The authors make some very general pitches about the importance of their work, including some attempts at management recommendations. However, to simply state that the currently recognized management units are too broad is not sufficient. Some additional advice from the authors is needed. Related to this point, and number (2) above, what role might translocations and captive breeding play in long-term management? Recognizing contemporary disruptions to continuity of populations will be an important part of decision making in this regard, so the authors need to provide more focused comments about these topics in the Discussion.

We look forward to receiving your revised manuscript.

Kind regards,

Christopher M. Somers

Academic Editor

PLOS ONE

Journal Requirements:

3. We note that Figure 1,  in your submission contain copyrighted images. All PLOS content is published under the Creative Commons Attribution License (CC BY 4.0), which means that the manuscript, images, and Supporting Information files will be freely available online, and any third party is permitted to access, download, copy, distribute, and use these materials in any way, even commercially, with proper attribution. For more information, see our copyright guidelines: http://journals.plos.org/plosone/s/licenses-and-copyright.

Reviewers' comments:

Reviewer's Responses to Questions

**Comments to the Author**

1. Is the manuscript technically sound, and do the data support the conclusions?

Reviewer #1: Yes

Reviewer #2: Yes

2. Has the statistical analysis been performed appropriately and rigorously? 

Reviewer #1: Yes

Reviewer #2: Yes

3. Have the authors made all data underlying the findings in their manuscript fully available?

Reviewer #1: Yes

Reviewer #2: Yes

4. Is the manuscript presented in an intelligible fashion and written in standard English?

Reviewer #1: Yes

Reviewer #2: Yes

5. Review Comments to the Author

Reviewer #1: In the study titled “Limited gene flow and pronounced population genetic structure of Eastern Massasauga (Sistrurus catenatus) in a Midwestern prairie remnant” Anthonysamy et al. explore patterns of fine-scale genetic diversity, genetic structure, and gene flow within and among threatened Eastern Massasauga individuals (n = 327), hibernacula (n = 9), and study areas (n = 3) in Illinois, U.S., using 21 independent nuclear microsatellite loci. In general, Anthonysamy et al. found that genetic connectivity among study areas—and even among hibernacula within study areas—is limited, which is congruent with previous studies performed on the same species at both broad and small spatial scales. Moreover, Anthonysamy et al. found that each study area is characterized by having <50 effective individuals and that genetic diversity within these study areas is moderate, which is also congruent with previous studies.

The study is well written and it is imperative to acknowledge that authors collected hundreds of valuable samples spanning 16 years, which must have presented itself as an insurmountable task at times. As such, the present study would be a nice addition to the PLOS ONE conservation genetics repertoire, although it is also true that this study carries a few limitations that should be addressed in the Discussion and/or that should demand reanalysis.

Major comments:

-Ideally, the present study should include less anthropogenically disturbed study areas (e.g., other Eastern Massasauga sites, perhaps from Canada) or a sister taxon (e.g., non-threatened Western Massasaugas) as a control to strengthen the author’s empirical case of limited gene flow among Eastern Massasauga hibernacula and study areas in Illinois due to recent anthropogenic effects. In other words, authors should present an “expected” genetic structure metric between undisturbed study areas, which should then be compared with the genetic structure metrics (from disturbed study areas) of this study. At the same time, it must be recognized that it is impossible to include less anthropogenically disturbed areas in this study just as it must be recognized that the use of microsatellite markers complicates cross-study comparisons that include anthropogenically undisturbed samples. Hence, this study may benefit from performing analyses in programs such as VORTEX (https://scti.tools/vortex/) or EASYPOP (https://www.unil.ch/dee/en/home/menuinst/open-positions-and-public-resources/softwares--dataset/softwares/easypop.html) where gene flow between undisturbed study areas may be simulated and from which “control” genetic structure metrics may be obtained.

-Please present a Supplemental Table or an Excel Spreadsheet with the individuals collected, locations, and dates of collection. Assuming a conservative generation time of 3 years for Eastern Massasaugas, the samples collected in this study span 6 generations on average. It is possible that, in some cases, significant genetic structure derives from temporal discordance (e.g., all samples from one area were collected in 1999 and all samples from another area were collected in 2015, to cite an extreme scenario). If this is the case, it would be recommended to subdivide each area or hibernaculum by time. This approach may even be useful for presenting a case of increased genetic isolation through time (if any) and it may or may not highlight the need for urgent management action.

-Please use G’’ST (as implemented in GenAlEx) instead of FST for pairwise comparisons using microsatellite markers (see https://onlinelibrary.wiley.com/doi/full/10.1111/j.1755-0998.2010.02927.x).

-In lines 269-270, please discuss in further depth why no significant inbreeding was found in each area and hibernaculum despite limited gene flow and small effective sizes. For instance, a recent study (https://onlinelibrary.wiley.com/doi/full/10.1111/mec.16147) found significant inbreeding (high FROH) in Eastern Massasaugas (including South Shore State Park) when compared to Western Massasaugas.

Minor comments:

-In line 122 please mention the criteria used to designate the three major study areas. For instance, looking at Fig. 1 it seems like South Shore State Park and Dam East and West Recreation areas form a continuum of suitable habitat.

-Please place hibernacula locations in Fig. 1 and present a “zoom out” map of the area.

-In line 173, an isolation-by-distance pattern implies that a formal Mantel test or a spatial autocorrelation test (as implemented in GenAlEx) was performed. If this was not the case, please consider removing such a conclusion from the text.

Reviewer #2: Review for PONE D-21-33512

In this manuscript the authors conducted a series of genetic analyses to evaluate the genetic structure and connectivity within a population of federally threatened snake, the eastern massasauga, in Illinois, USA, using microsatellites from samples collected over a 16-year period. The found strong evidence of population structure between three sampling areas with weaker evidence of substructure within populations. The lack of connectivity among populations was attributed to both contemporary (i.e., anthropogenic) and historical habitat fragmentation and high spatial fidelity and low movement potential of this species.

I thought this manuscript was very well structured and organized and the analyses and conclusions sound. In fact, I have no major comments for the authors and recommend that this manuscript be accepted with minor revisions. I appreciated how the authors made the point that the current three conservation units for eastern massasauga are overly broad and do not account for more localized conservation and recovery needs. I think the information in this manuscript provides important information for the management and recovery of this population of eastern massasauga.

Minor Comments:

The authors mention that the snakes at this study site over-wintered in crayfish burrows. I suspect that crayfish burrows could be very dispersed across mesic habitats so how did the authors group crayfish burrows into different hibernaculum? Were the crayfish burrows used by the snakes more spatially clustered? Some clarification on this would help interpret the results of within-population analyses (e.g., Figure 3).

It would be helpful to see some diagnostic plots for selecting K from the STRUCTURE analyses, either plots of the mean posterior probability of the data (LnP(K)) or Delta K as a function of K. Furthermore, the authors might consider the method proposed by Puechmaille (2016) for identifying the optimal value of K when Delta K suggests K=2. This approach can be implemented using the online tool STRUCTURESELECTOR.

Puechmaille, S. J. 2016. The program STRUCTURE does not reliably recover the correct population structure when sampling is uneven: subsampling and new estimators alleviate the problem. Molecular Ecology Resources 16:608–627.

Li, Y. L., and J. X. Liu. 2018. STRUCTURESELECTOR: a web‐based software to select and visualize the optimal number of clusters using multiple methods. Molecular Ecology Resources 18:176–177.

This may be beyond the scope of the study, but since the authors claim demographic independence among their three populations I was curious as to how demographic parameters actually vary among those populations. The data may not be available (or sample sizes insufficient) but, for example, do they see differences in survival or fecundity among those populations?

Line 424: Should this be “The number of effective migrants….”

Tables 1 and 3: What do the numbers “1” represent in the above-diagonal part of these tables? Should they be removed? Also, the font size within the tables varies; this should be changed to be the same throughout the table.

6. PLOS authors have the option to publish the peer review history of their article (what does this mean?). If published, this will include your full peer review and any attached files.

Reviewer #1: No

Reviewer #2: No

---

## [Author Response · Author response to Decision Letter 0]

23 Feb 2022

We have made the following required changes:

1. Updated Figure 1 to include a larger geographic context for the study site. 

2. Expanded the discussion to provide the readers with more guidance about historical vs. contemporary fragmentation as well as the effects of fragmentation. Sections where this has been addressed include in the Discussion under the headings, Population structure and gene flow (starting at line 271) and Genetic diversity (starting at line 305). We did not incorporate and additional analysis (dataset simulation) as suggested by Reviewer 1 as we feel that we have addressed the concern to include a undisturbed population with changes to the text. For example, we discuss how it can be difficult to separate the effects of historical and contemporary landscape disturbance when natural features coexist with historic ones. Further, we acknowledge that recent genomic work (Ochoa and Gibbs, 2021) suggests that Eastern Massasauga has persisisted historically (10,000 ybp) at low Ne which could lead to the argument that we might expect strong population structure. However, we also point out that the same genomic study also detected recent population bottlenecks corresponding to anthropogenic fragmentation. This study also detected high levels population inbreeding that result from small population size and and a lack of natural selection against moderately deleterious alleles, which will exacerbate threats to long-term persistence. 

3. We have provided more focused comments throughout the Discussion about the impacts of anthropogenic/contemporary disruptions (see #2 above). Additionally, under the Discussion heading: Implications for recovery of Eastern Massasauga (starting at line 329), we added the following sentence at the very beginning of the section: “Anthropogenic fragmentation can erode genetic diversity in vulnerable Eastern Massasauga populations as contemporary barriers to dispersal prevent gene flow, increase inbreeding depression, reduce adaptive variation, thereby accelerating populations into an extinction vortex (5,7,8).” We have also elaborated on management recommendations, citing previous studies that have made ecological-related management recommendations, and have added text to provide more explicit management recommendations with regard to genetic rescue (translocations and captive rearing).

We have addressed the following:

Reviewer #1:

Major comments –

• Re: including less anthropogenically disturbed areas or “expected” genetic structure. See comments above for required change #2. 

• We have provided a supplemental excel files that includes genotypes, location and year of collection for each sample used in the analyses. For the most part, samples were collected throughout the duration of the study. In addition, we have shown that there little temporal change in genetic diversity over a 10-year period in samples analyzed from South Shore State Park (Baker et al. 2018). In short, we do not expect our observed genetic structure to correspond to temporal discordance. We have acknowledged this concern in the Results (lines 193-195).

• We have now included the unbiased estimator, G’’ST, in our pairwise comparison of genetic structure, but have also retained our estimates of FST to make results comparable to previously published studies. See lines 141-142 in the Methods, lines 177-178 in the Results as well as Tables 2 and 3. 

• We have expanded the discussion to elaborate on why no significant inbreeding was found in each area and hibernaculum despite limited gene flow and small Ne. We cite the Ochoa and Gibbs 2021 study that did detect inbreeding (FROH) at SSSP and we acknowledge the potential limitations of using microsatellite markers to detect inbreeding when He is moderate and Ne has been historically small (See lines 315-325).

Minor comments – 

• We have included a brief description of how hibernacula were designated. “The primary study areas are separated by distances of ~3-5 km and hibernacula (contiguous low-lying mesic grasslands with low canopy cover) within study areas are separated by unsuitable habitat and anthropogenic boundaries from a few kilometers to a few hundred meters.” Lines 123-125. 

• We have updated Fig 1 to present a zoomed out map of the area but we did not add specific hiberacula locations to protect the populations from collection and persecution. Lines 125-126.

• We have removed isolation-by-distance text. Line 184 in the tracked changes version.

Reviewer #2

Minor Comments – 

• Re: distribution of crayfish burrows: Hibernacula were determined by contiguous suitable habitat, low lying wet grasslands with low canopy cover. Therefore, hibernacula represent grassland patches across the landscape separated by various habitat (woodland, shrub-scrub, river, pond/lake etc.) and anthropogenic (roads, trails, rail lines, urbanization etc..) boundaries. Thus, hibernacula are separated by at least one type of barrier but permeability of movements across barriers differs (snakes will move through woodlands more readily than across roadways). Yes, even within a grassland habitat patch, crayfish burrow density differs depending on how mesic the microhabitat is. By default, areas with more crayfish burrows have more Massasaugas hibernating in them. We have provided a better description of how hibernacula were designated in the methods. (See lines 123-125).

• We greatly appreciated the suggestion to use the method of Puechmaille (2016) and Structure Selector (Li and Liu 2018) to identify the optimal value of K and have implemented this approach in our revised version of the manuscript. See lines 143-151 in the Methods, lines 181-193 in the Results as well as updated Fig 2-3 and the addition of two new tables (Table 5-6) that compares the values of K across stepwise coefficient membership thresholds. We have also included all Structure Selector output plots in our Supplementary files.

• Unfortunately, we do not have estimates of demographic parameters (e.g., survival, fecundity, etc.) for the three populations. These have been estimated for SSSP (see Sarah Baker’s dissertation; Baker 2016), but sufficient numbers of recaptures are lacking for EHSP or DMRA to calculate robust estimates. 

• Line 424: We double-checked the title of the article and it is correct. “Detecting the number of clusters of individuals using the software STRUCTURE: A simulation study”

• Tables 1 and 3 have been updated. The number “1” was a simple formatting issue that has been corrected. 

We have also made the following changes to address the journal requirements:

1. Updated the formatting for the main body and title page to meet PLOS ONE style requirements. Updates are highlight in the marked version of the manuscript. 

2. Provided an excel file (S1) of the raw genotype data for all samples used in analyses. This file also includes collection year and hibernaculum abbreviation for each sample as requested by the reviewers.

3. Updated the legend for Figure 1 (Study site map) to include the following statement: All map data (road layer, municipal boundaries, streams, lakes, and protected lands) were publicly accessible and downloaded from the Illinois Geospatial Data Clearinghouse (https://clearinghouse.isgs.illinois.edu/data) and from I-View (https://www.prairiestateconservation.org/pscc/iview/) and then projected in ARCMAP 10.8.0 (ESRI, Redlands, CA).

---

## [Editor Report · Decision Letter 1]

7 Mar 2022

Limited gene flow and pronounced population genetic structure of Eastern Massasauga (Sistrurus catenatus) in a Midwestern prairie remnant

PONE-D-21-33512R1

Dear Dr. Anthonysamy,

We’re pleased to inform you that your manuscript has been judged scientifically suitable for publication and will be formally accepted for publication once it meets all outstanding technical requirements.

Kind regards,

Christopher M. Somers

Academic Editor

PLOS ONE
---

## [Editor Report · Acceptance letter]

11 Mar 2022

PONE-D-21-33512R1 

Limited gene flow and pronounced population genetic structure of Eastern Massasauga (Sistrurus catenatus) in a Midwestern prairie remnant 

Dear Dr. Anthonysamy:

I'm pleased to inform you that your manuscript has been deemed suitable for publication in PLOS ONE. Congratulations! Your manuscript is now with our production department. 

Kind regards, 

on behalf of

Dr. Christopher M. Somers 

Academic Editor

PLOS ONE